# Prototyping in Polymethylpentene to Enable Oxygen-Permeable On-a-Chip Cell Culture and Organ-on-a-Chip Devices Suitable for Microscopy

**DOI:** 10.3390/mi15070898

**Published:** 2024-07-10

**Authors:** Linda Sønstevold, Paulina Koza, Maciej Czerkies, Erik Andreassen, Paul McMahon, Elizaveta Vereshchagina

**Affiliations:** 1Department of Smart Sensors and Microsystems, SINTEF Digital, Gaustadalléen 23C, 0373 Oslo, Norway; 2Institute of Fundamental Technological Research, Polish Academy of Sciences, Pawińskiego St. 5B, 02-106 Warsaw, Poland; 3Department of Materials and Nanotechnology, SINTEF Industry, Forskningsveien 1, 0373 Oslo, Norway; erik.andreassen@sintef.no (E.A.);

**Keywords:** polymethylpentene (PMP), oxygen control, gas permeability, organ-on-a-chip, microscopy, prototyping thermoplastics, microfluidic device

## Abstract

With the rapid development and commercial interest in the organ-on-a-chip (OoC) field, there is a need for materials addressing key experimental demands and enabling both prototyping and large-scale production. Here, we utilized the gas-permeable, thermoplastic material polymethylpentene (PMP). Three methods were tested to prototype transparent PMP films suitable for transmission light microscopy: hot-press molding, extrusion, and polishing of a commercial, hazy extruded film. The transparent films (thickness 20, 125, 133, 356, and 653 µm) were assembled as the cell-adhering layer in sealed culture chamber devices, to assess resulting oxygen concentration after 4 days of A549 cell culture (cancerous lung epithelial cells). Oxygen concentrations stabilized between 15.6% and 11.6%, where the thicker the film, the lower the oxygen concentration. Cell adherence, proliferation, and viability were comparable to glass for all PMP films (coated with poly-L-lysine), and transparency was adequate for transmission light microscopy of adherent cells. Hot-press molding was concluded as the preferred film prototyping method, due to excellent and reproducible film transparency, the possibility to easily vary film thickness, and the equipment being commonly available. The molecular orientation in the PMP films was characterized by IR dichroism. As expected, the extruded films showed clear orientation, but a novel result was that hot-press molding may also induce some orientation. It has been reported that orientation affects the permeability, but with the films in this study, we conclude that the orientation is not a critical factor. With the obtained results, we find it likely that OoC models with relevant in vivo oxygen concentrations may be facilitated by PMP. Combined with established large-scale production methods for thermoplastics, we foresee a useful role for PMP within the OoC field.

## 1. Introduction

Organ-on-a-chip (OoC) technology promises to revolutionize biomedical research and drug development by providing more accurate and ethical human models for testing the efficacy and safety of drugs and treatments [1,2,3]. By relying on the new research on mechanisms of drug resistance and transport, as, e.g., reported for the liver [4], and combined with the accurate replication of the cell microenvironment, OoC models may become an important tool in preclinical assessments of the efficacy and safety of new drugs, as well as potentially contribute to drug optimization. The selection of appropriate materials for OoC and microfluidic devices, which shall sustain long-term cell culture experiments, is crucial for the development of successful and relevant physiological models [5,6]. Among the criteria for the materials in cell culture devices, oxygen permeability is considered one of the most important [7]. The oxygen concentration strongly affects the viability and function of cultured cells, organoids, and tissues. Specifically, the oxygen level profoundly influences cell metabolism, proliferation, differentiation, and, eventually, death [8]. Insufficient oxygen supply can trigger pathological processes in cells [9], consequently putting under question the relevance of model-to-in vivo conditions, leading to inconsistent results and compromising reproducibility. On the contrary, adequate oxygen levels allow cells to differentiate normally and, under otherwise favorable conditions, to develop predictably, enabling reliable on-a-chip studies [10]. With appropriate oxygen supply, the microfluidic environment more closely mimics the natural tissue environment, resulting in minimal or no compromise to cell metabolism [11]. An important factor to keep in mind when developing OoC models is that the oxygen concentration in in vivo organs does not match that of ambient air [12,13,14]. As oxygen concentration differs between in vivo organs, “adequate” oxygen levels in OoC models will vary with the organ of interest. To successfully enable OoC models, the development of materials and devices that accurately monitor and modulate the oxygen gradient within the microfluidic devices is therefore crucial, covering the oxygen levels from that in arterial blood (13–14%) to the core of hypoxic tumors (1–2%) [12,15]. This is particularly important, for example, in assessing the efficacy and toxicity of drugs on cells and organ models cultured within microfluidic environments, emphasizing the importance of research towards new OoC materials and devices for the pharmaceutical and biotechnology industries [16]. The emergence of in situ oxygen measurement techniques responds to the urgent need to control oxygen levels in OoC models [17].

In the context of biological experiments, besides gas permeability, other requirements for the materials are (i) biocompatibility, (ii) suitability for microscopy analysis of cells (transparency, first of all), (iii) chemical compatibility with a given cell assay or other reagents, (iv) mechanical robustness, (v) compatibility with common sterilization methods, and (vi) various aspects of manufacturability (fabrication cost, potential for upscaling to high volume, feasibility of micro- and nanostructuring, and potential for integration of sensors and other materials, in general).

Polydimethylsiloxane (PDMS) is widely applied for OoC and long-term cell culture on-a-chip devices due to its excellent biocompatibility, optical transparency, gas permeability, and flexibility, all essential for mimicking physiological conditions and observing cellular behaviors [18,19]. The ease of fabricating prototype devices in PDMS for various applications [20], including cellular studies, results in a low threshold for many researchers when entering the field and allows for a short design screening phase, thus making this material a popular choice for the development of cell-friendly microenvironments. However, PDMS also has some drawbacks, mainly its tendency to absorb small hydrophobic molecules which may interfere with drug studies and its low mechanical robustness, limiting its application in miniaturized cell culture devices [21].

Thermoplastic polymer materials, such as polystyrene (PS) and polymethyl methacrylate (PMMA), have also been used [22]. While thermoplastic materials often offer cost-effective production, optical transparency, and superior mechanical strength compared to PDMS [23], these materials generally have lower gas permeability and, in many cases, require surface treatment to improve biocompatibility and wettability or to hinder non-specific adsorption. Glass as a material has many excellent qualities for biological experiments, e.g., transparency, inertness, and durability, but it is also more expensive than polymers, more brittle than the materials described above, and requires specialized fabrication methods for the development of miniaturized devices [24]. Another important group of materials is hydrogels. Many studies have demonstrated their excellent properties for constructing soft tissue models, specifically biocompatibility and mechanical properties closely resembling natural cell growth environments [25]. However, their unique chemical and physical properties come at the cost of multiple challenges in controlling them over time and addressing them individually.

The development of bio-fabrication methods for artificial, biomimetic structures has advanced rapidly in recent years [26]. Various curable biocompatible materials as well as biomaterials have been successfully combined in sophisticated three-dimensional (3D) networks for OoC applications, layer-by-layer, using 3D (bio)printing technologies [27,28,29].

The present work focuses on the material polymethylpentene (PMP). PMP has previously drawn attention as a membrane material for blood oxygenation systems and, alongside commonly used PDMS, has a potential for the development of microfluidic artificial lungs [28]. We have previously shown that PMP fulfills the multiple requirements for materials used in OoC devices, being a thermoplastic with unique gas permeability properties [30]. None of the above discussed materials, including PDMS, can offer such a unique combination of properties, to the best of our knowledge. Surprisingly, in the context of microfluidic devices designed for long-term culturing of cells and OoC, PMP has been relatively little studied up to now [30,31,32]. We believe that PMP has a large potential in these applications. Recent studies evaluating PMP as an alternative to PDMS and conventional tissue culture polystyrene (TCPS) for cell culture also report on its superior properties [33]. However, to fully uncover the potential of PMP, more research is required on all aspects of prototyping of miniaturized devices, as well as application-specific oxygen supply performance.

When it comes to the fabrication of OoC devices, one of the major reasons why PDMS has become so popular in the OoC community is the ease of device prototyping [34]. However, challenges in upscaling and low fabrication throughput have restricted the large-scale production and successful commercialization of PDMS-based devices, e.g., for the pharmaceutical and biotechnology industries [35]. For thermoplastics, such as PMP, large-scale production methods are well established, with techniques such as injection molding for making 3D parts and various methods for making films and sheets. However, commonly available prototyping methods for thermoplastics, such as milling, laser ablation, and 3D printing, typically do not yield the low surface roughness required for adequate optical transparency for cell imaging, although post treatment such as mechanical or chemical polishing might give the required low roughness. To achieve the optical surface finish, prototyping should be performed with methods that employ two polished forming surfaces (using a mold, rolls, etc.), e.g., injection molding, hot embossing, or certain thermoforming methods [36].

Transparent PMP films were prototyped in this study. Prototyping of films is usually not a part of OoC fabrication studies. However, the fabrication of devices by stacking and bonding structured films and sheets (e.g., structured by laser cutting) has been reported [37]. Such studies utilize (unstructured) films acquired from commercial suppliers.

Three film prototyping methods were used in this study: film extrusion, hot-press molding, and polishing of a commercial film with one frosted (rough) side (this commercial film was also extruded). Film extrusion processes range from large-scale industrial production lines to lab-scale film extrusion lines, such as the one used in this study. There are several studies of PMP cast film extrusion [38,39] and blown film extrusion [40,41]. The second process in our study, hot-press molding, can be performed with straightforward and commonly available equipment, and special pressing tools are available for prototyping films with different thicknesses for both research and development purposes. Tian et al. [42] pressed PMP films as thin as 20 μm. To achieve this, the material was melted and pressed between two Al foils, which were later removed by dissolving them in a NaOH solution. Finally, polishing a commercially available film is the simplest of the three methods in this study. Film polishing was also used in our previous study [30]. Polishing can be performed manually or semi-manually.

This work is a follow-up of our previous study [30], which focused on introducing the potential of PMP for cell studies on-a-chip over several days. The previous study compared 125 µm thick PMP to PDMS and glass. This paper reports on the prototyping of transparent PMP films with thicknesses in the range 20 to 653 µm by three different prototyping methods (extrusion, hot-press molding, and polishing of a commercial film) and their integration into microfluidic devices to assess the supply of oxygen to a monolayer of cancerous lung epithelial cells. Assessing the effect of film thicknesses on the oxygen level enables informed decisions on device design and fabrication alternatives, as different fabrication methods are compatible with different device wall thicknesses. Based on the resulting oxygen concentrations, the feasibility of PMP devices to replicate in vivo oxygen concentrations is discussed. The effects of prototyping method on cell adherence, morphology, proliferation, viability, and microscopy imaging are also assessed. Furthermore, we report on our experience with the selected prototyping methods and advise on the selection of prototyping technique in different circumstances. Compared to earlier studies using PMP [33,43,44], which mainly focused on cell culture plate formats, this contribution highlights the potential for prototyping miniaturized microfluidic and OoC devices in PMP, with oxygen supply solely through the material itself, for long-term cell culture studies. We believe that considering the high growth rate of the OoC field [45,46] and the continuous evolution of new quantification technologies on the cellular level, which provide essential input for developing more accurate OoC [47,48], the need for innovative materials addressing the variety of experimental demands is evident and growing. The important interaction between the fields of material science and microfluidics is increasingly recognized by the OoC and microfluidics communities [49], and we also observe the initial adoption of these technologies by industry. The present work is an attempt to further respond to this challenge.

## 2. Materials and Methods

### 2.1. Design of the Culture Chamber Devices

The devices (Figure 1a) consisted of four assembled layers, from layer 1 representing the cell culture bottom to layer 4 representing the lid of the device. There were two variants of the lid, resulting in two variants of the device—referred to as “*Sensor*” and “*Non-sensor*” lid/device, see Figure 1a. Layers 1 to 3 were identical for both variants. Each device contained two cell culture chambers which served as two technical replicates in cell experiments. Layer 1 was a transparent film of PMP with thickness 20–653 µm. As a gas-impermeable control, devices were made with a glass cover slip as layer 1. Layer 2 was a pressure-sensitive adhesive (PSA), adhering layer 1 to layer 3 (the baseplate). The PSA was cut to fit the diameter of the culture chambers. The baseplate (layer 3) was milled in polycarbonate (PC) and housed the two cell culture chambers with a diameter of 6.4 mm and a height of 4 mm. The chamber diameter was selected to be equal to that in regular 96-well plates [50]. The lid (layer 4) was milled in PC. In the Sensor device (Figure 1a to the left, Figure 1b,d), the purpose of the lid was to provide a tight seal for the culture chamber with integrated oxygen sensing. The lid, with a cylindrical cavity (ID 10 mm, OD 13.6 mm, depth 1.2 mm) to insert a 10 × 1.5 mm^2^ silicone O-ring (Otto Olsen, Skedsmokorset, Norway), sealed the device tightly when secured by screws. Spots for oxygen sensing, SP-PSt3-NAU-D3-YOP, with diameter 3 mm (PreSens, Regensburg, Germany) were positioned centrally and assembled with lids by gluing with a silicone glue Dowsil SG2 734 (PreSens, Regensburg, Germany). Sockets in circular shape, distanced 1 mm from the spots used for oxygen sensing, were designed to accommodate an optical fiber for optical readout as illustrated in Figure 1b,d. In the Non-sensor device (Figure 1a to the right and Figure 1c), the lid was only a cover to ensure sterility.

### 2.2. Fabrication Methods of the Culture Chamber Devices

#### 2.2.1. Prototyping and Characterization of Transparent PMP Films—Layer 1

Five different transparent PMP films were compared in this work, see Table 1. These were made with the following prototyping techniques: (i) extrusion (PMP-E-20), (ii) mechanical polishing of a commercial film (PMP-P-125), and (iii) hot-press molding (PMP-M-133, PMP-M-356, PMP-M-653).

For the extrusion (Figure 1e) and molding (Figure 1f), the feedstock was granulates of the PMP grade TPX DX845 from Mitsui Chemicals (acquired from Goodfellow, Huntingdon, UK). According to the material producer, this grade is a copolymer of 4-methyl-1-pentene and 1-decene.

The extruded film PMP-E-20 was made with a lab-scale cast film extrusion line (Collin Lab & Pilot Solutions, Maitenbeth, Germany) consisting of an extruder (E 20T) and a take-up device (CR 72T). The melt temperature was 280 °C, and the extruder screw rotation speed was 50 rpm.

The hot-press-molded films were made with a manual press with 21 × 21 cm^2^ steel platens (Fontijne Presses, Rotterdam, The Netherlands). PMP granulates were placed between glass plates (17 × 17 × 0.3 cm^3^). The gap between the glass plates was defined by shims (i.e., spacers). The assembly was heated and pressed. The temperature of the metal platens was 247 °C. The assembly was heated for 20 min for the thinnest film and 70 min for the thickest film. After heating, the pressing time was 5 min for all film thicknesses, and the pressure was optimized for each thickness (0.45 to 0.90 MPa).

For the polished film PMP-P-125 (Figure 1g), the starting point was a 125 μm thick commercial film (ME31110, Goodfellow, Huntingdon, UK). One side of the film had high surface roughness. Hence, the film was hazy, and the rough side had to be polished to achieve optical transparency and clarity. According to the supplier, this film was made of the same PMP grade as above (TPX DX845). The polishing was performed with a polishing kit for acrylics (PMMA) (Acrylic Scratch Remover, Quixx System, Garasjetid, Oslo, Norway), containing sandpaper, polishing paste, and cotton cloths. The film was placed on a silicone mat and sanded down with sandpaper (grit 1500), before the cotton cloth was used to polish with the polishing paste, first in the vertical direction, spending 3 min per vertical stripe, and thereafter in the horizontal direction, spending 3 min per horizontal stripe. The polishing paste was wiped off with isopropanol and DI water.

The PMP films were cut into 2 × 4 cm^2^ pieces using scissors and rinsed in acetone, then isopropanol, then DI water, before they were blown dry with nitrogen gas.

The molecular orientation in the PMP films was characterized by IR spectroscopy (Agilent Cary 670 FTIR Spectrometer, Santa Clara, CA, USA). The measurements were performed in transmission using a wire grid polarizer (PerkinElmer, Waltham, MA, USA) in the beam path, thereby characterizing the polymer chain orientation by IR dichroism measurements. Dichroic ratios were calculated with modules in the SciPy Python library (version 1.11.4). First, the IR absorption spectra were smoothed with the Savitzky–Golay method, and the second derivatives of the spectra were calculated. Cubic splines were then fitted to the second derivatives. For each spline (spectrum), the amplitude and position of 10–12 relevant absorption bands were determined. The dichroic ratio for a given band was then obtained as the ratio of peak amplitudes measured with parallel and perpendicular polarization.

#### 2.2.2. Fabrication of Layer 2–4 and Assembly

For layer 2, PSA 92712 (Adhesives Research, Arcare^®^, Limerick, Ireland) was cut with a scalpel to fit the size of the PMP film or glass coverslip, and 6.4 mm holes were cut to fit the chamber diameter. Layers 3 and 4 (the baseplate and lids) were milled in PC (Lexan, Astrup AS, Oslo, Norway) using a 3-axis milling machine (DMG DMC 1035 V, 10,000 rpm, rough/fine milling performed using 3/0.5 mm endmill, 500/300 mm/min feed rate, 3/0.05 mm axial depth of cut and 0.8/0.2 mm radial depth of cut). After milling, the parts were wiped with isopropanol, quickly rinsed in DI water, and exposed to ultrasonication in DI water for 5 min. For the Sensor lids, the spots for oxygen sensing were assembled with the lids manually by gluing and left to dry for 24 h at room temperature. Eighteen sensors in total were characterized in air. A mean oxygen concentration of 21.5% with standard deviation of 0.03% was found. O-rings were inserted into the cavities.

For assembly, one of the five prototyped PMP films described above (Section 2.2.1) or a glass coverslip (Paul Marienfeld GmbH & Co, Lauda-Königshofen, Germany, #0101060) (thickness no. 1; 0.13–0.16 mm) was attached to a baseplate using the cut PSA. In the remainder of the paper, these assembled layers (1–3) will be referred to as “the base”. During experiments, the *Non-sensor lids* were simply placed on top of the baseplate (Figure 1c), while the *Sensor lids* were positioned and fastened tightly by two screws (Figure 1b).

### 2.3. Cells and Culture Conditions

An adenocarcinomic human alveolar epithelial cell line A549 was sourced from American Type Culture Collection (ATCC, Manassas, VA, USA, cat. CCL-185). The cells were cultured in F12K medium (Thermo Fisher Scientific, Waltham, MA, USA, cat. 21127030), which was supplemented with 10% fetal bovine serum (Thermo Fisher Scientific, Waltham, MA, USA, cat. A3160802) and penicillin/streptomycin antibiotic solution (Thermo Fisher Scientific, Waltham, MA, USA, cat. 15140122). ThermoFisher series 8000 WJ incubator and standard conditions (37 °C, 5% CO_2_) were used for cell culture. Cells were passaged every 2 to 3 days once they reached 90% confluence.

### 2.4. Comparison of Oxygen Concentration in Sensor Devices with Different PMP Prototypes

Sterilization of assembled devices was performed as follows: The base and lids (both Sensor and Non-sensor lids) were soaked in 70% ethanol for 30 min. In addition, the base part was exposed to UV light for 30 min. Sterile PBS was used for washing of both parts (Thermo Fisher Scientific, Waltham, MA, USA, cat. 20012-027), followed by drying in air in a laminar hood. Culture chambers were coated with 0.01% solution of poly-L-lysine (Sigma-Aldrich, Burlington, MA, USA cat. P4707) for 10 min and then washed with PBS, prior to cell seeding. TrypLE reagent was used to detach A549 cells from culture flasks (Thermo Fisher Scientific, Waltham, MA, USA, cat. 12604013). The cells were further counted using a TC20 Automated Cell Counter (Bio-Rad Laboratories, Hercules, CA, USA). Before covering the bases with Non-sensor lids, the cells were seeded at a density of 20,000 cells per culture chamber in the full culture medium. Cells were permitted to adhere overnight under typical culture conditions. At the beginning of the experiment, the medium was aspirated from the cells, and 250 µL of fresh medium, pre-heated to 37 °C, was added to make sure the initial conditions were the same for all experimental variants. Right after medium exchange, the Sensor lids were placed onto the bases and tightly secured. The devices were transferred into the incubator and permitted to equilibrate to 37 °C for an hour prior to the first measurement. To exclude any disturbances from the air exchange through the bottom of the device, the devices were kept suspended 1 cm above the incubator shelf.

To carry out oxygen measurements, the devices were taken out from the incubator and placed on the hot plate set for 37 °C (Figure 1b, to the left) for about 1 min. This was to prevent the devices from cooling and to equilibrate them to the same temperature.

All measurements were carried out using an Oxy-1 SMA oxygen meter (PreSens, Regensburg, Germany) combined with an optical fiber (PreSens, Regensburg, Germany, POF-L2.5-2SMA). Data were collected using PreSens Measurement Studio 2 software (version 24.0.0.2293). For each cell culture chamber, five measurements were taken per time point using the following settings: temperature—37 °C; pressure—1025 hPa; mode—humid; and salinity—10 pmil. The median of the five oxygen measurements was extracted for each culture chamber and time point. The average of the two technical replicates per biological replicate was identified. It was further used to calculate the average and standard deviation of the oxygen measurements for all biological replicates. There were four biological replicates in total.

### 2.5. Comparison of Cell Adherence, Proliferation, and Viability on Different PMP Prototypes

The devices were assembled and sterilized as described in Section 2.4, with Non-sensor lids. In addition, one hot-press-molded PMP film was plasma treated prior to sterilization. A low-pressure plasma cleaner equipped with a 13.56 MHz/50 W generator (Zepto, Diener electronic GmbH, Ebhausen, Germany) was used, and the procedure was as follows: (1) air was evacuated from the working chamber, and pressure was reduced to 0.1 mbar; (2) oxygen was allowed to the working chamber (5 min, flowrate 10 sccm); and (3) the plasma treatment process was performed (50 W, 5 min) at an oxygen flowrate of 10 sccm.

Subsequently, cells were seeded into culture chambers using the same method outlined in Section 2.4, i.e., at a density of 20,000 cells per cell culture chamber. Non-sensor lids were placed on the devices, and they were transferred into incubators. Cells were permitted to adhere during 6 h at 37 °C prior to taking the devices out of the incubator and transmitting them to the Leica SP5 confocal microscope equipped with environmental chamber, ensuring the same culture conditions (37 °C and 5% CO_2_). Transmitted light images of living cells were taken with differential interference contrast (DIC) at the 6, 24, and 48 h time points. At least three fields of view were captured for each culture chamber.

At the conclusion of the experiment, lids were taken off, and the medium was aspirated. Calcein-AM (Sigma-Aldrich, Burlington, MA, USA, cat. 56496) was diluted 1:2000 in a serum-free F12K medium, consequently added to the cells for 15 min and then aspirated. Following three times of washing with a full culture medium, cells were immediately analyzed for fluorescence, indicating cell viability, under a Leica SP5 microscope. All images were captured using an HCPL APO 20×/0.70 objective (Leica Microsystems, Wetzlar, Germany) and Leica LAS AF software (version 2.7.3.9723).

## 3. Results

### 3.1. Prototyping and Characterization of Transparent PMP Films

An overview of the PMP films prototyped in this work is shown in Table 1. Photographs of the films showing their appearance at the macroscale are shown in Appendix A.

Prototyping of PMP films by extrusion yielded highly transparent films. This is demonstrated in Appendix A where the pattern of the background surface is clearly visible through the film. However, at the macroscale, the film was slightly wavy, as seen in Appendix A, making it difficult to cut areas of 2 × 4 cm^2^ with a planar surface optimal for cell imaging. The waviness was probably caused by a somewhat unstable extrusion process. With more material available, it should be possible to establish a process with adequate stability. A different PMP grade may also have a rheology which is more optimal for this film extrusion line.

Polishing the commercial non-transparent extruded film provided a significant increase in transparency and clarity, as seen in Appendix A, with the polished film to the left and the unpolished film to the right. Although transparency was significantly improved, the film had slight remnants of the texture of the original film after polishing, and during polishing, some small dents in the film appeared. The film was planar in its original condition and remained so after polishing, as seen in Appendix A. The polishing procedure was easy to perform but time-consuming and labor-intensive.

The hot-press-molded PMP films were transparent and had even thickness. They are shown in Appendix A. The pressing replicated the smooth surface of the glass plates, which is optimal for cell imaging. The film dimensions were about 8 × 8 cm^2^, and the films were often thinner towards the periphery and with a faint yellow color at the edges due to the thermo-oxidative degradation of the PMP. The thin and yellowish parts were cut away, and only the central parts of the films were used, with even thickness and no discoloring.

The hot-press molding procedure required some optimization to find the appropriate parameters for each film thickness (temperature, heating time, pressure, pressing time, and mass of granulates), in order to produce transparent films without marks from granulate boundaries not fully melted, while avoiding yellowing from thermo-oxidative degradation. However, once found, using those parameters enabled the reproducible manufacturing of films. With optimized parameters, the procedure was relatively easy to perform. Positioning the granulates on the bottom glass plate was hands-on and time-consuming, while the remainder of the procedure was time-consuming but mostly involving waiting time with a few manual operations.

Due to the nature of the simple experimental setup, in which the shims were sandwiched along the periphery of the glass plates without attachment, the shims were pressed out by the melted PMP mass when the glass plates were pressed together. This resulted in the PMP films being thinner than the thickness of the shims. However, the film thickness was adequately reproducible between replicates to provide, in total, six 2 × 4 cm^2^ samples from 2 to 3 molding replicates with a thickness standard deviation of less than 6% for all thicknesses: Films made with shims of thickness 150 µm, 400 µm, and 900 µm were measured to 133 ± 8 µm, 356 ± 11 µm, and 653 ± 15 µm, respectively, where the standard deviations are based on six samples as described above.

The preferred molecular orientation in the films, i.e., the anisotropy of the microstructure, was characterized by IR dichroism measurements. In the PMP literature, the absorption band at 918 cm^−1^ is often used for this purpose [38,51,52]. In this study, several other bands were also analyzed, with different angles between their dipole transition moment and the polymer chain axis. Some results are shown in Figure 2. The extruded and polished films (PMP-E-20 and PMP-P-125; the latter also extruded) had clear orientation, i.e., dichroic ratios significantly larger or smaller than 1. The thinnest hot-press-molded film (PMP-M-133) showed a low degree of orientation, while the two other hot-press-molded films were practically unoriented (the thickest film is not included in the figure). For some bands in the dataset, there were also shifts in wavenumber between peaks measured with the two polarizations. Films with high orientation had large shifts. The largest shifts (up to 2.8 cm^−1^) were observed for the 790 cm^−1^ band.

### 3.2. Evaluation of Oxygen Concentration in Sensor Devices with Different PMP Prototypes

As an introductory experiment, and to verify that the experimental set-up was working similarly as in previous studies [30], an experiment with six Sensor devices was performed—with glass in addition to the five different PMP prototypes as layer 1 (Figure 3, *n* = 1). The glass and PMP-P-125 samples link this study to our previous study [30], and the results for oxygen concentration were found to be comparable. Therefore, to reduce the workload, follow-up biological replicates of the experiment were performed with the five PMP prototypes only (Figure 4, *n* = 4).

The Sensor devices contained sealed cell culture chambers (Figure 1), where all but the bottom wall was made of PC, which has very low gas permeability [30]. Hence, our devices tested the oxygen permeability of the material in the bottom wall (layer 1) and its ability to supply the cultured A549 cells with oxygen during four days of cell culture. As expected, when glass was used as the bottom wall, the oxygen concentration dropped steadily within the first 24 h and stabilized at approximately 3% for the remainder of the experiment. For all the PMP prototypes, with thicknesses ranging from 20 µm to 653 µm, the oxygen concentrations stabilized between approximately 15.6% and 11.6%. The thicker the film, the lower the oxygen concentration inside the device. Both in the previous [30] and current study, oxygen concentrations stabilized at approximately 3% and 15–16% with glass and PMP-P-125 as layer 1, respectively. In the previous study [30], we also tested a 200 µm thick PDMS film, which also resulted in the oxygen concentration stabilizing at 16%. The theoretical value for oxygen concentration inside a cell culture incubator is 18.6% [12].

In the plot in Figure 4, the standard deviations are relatively large. This may be due to variations in exact cell seeding number, cell passage, incubator conditions, sensor reading errors, etc., as the plot depicts the average and standard deviation of the exact measured values, not normalized data. Although the standard deviations are relatively large, the trends are clear.

### 3.3. Evaluation of Cell Adherence, Proliferation, and Viability on Different PMP Prototypes

Transmission light images of A549 cells cultured in Non-sensor devices, taken at 6, 24, and 48 h, align with previously published results with the polished, commercial film (PMP-P-125) [30]. When coated with poly-L-lysine, all the tested PMP prototypes supported adherence of cells to a comparable degree to glass, as illustrated in Figure 5 and Figure 6. Also, for all PMP prototypes and glass, the cells exhibited the typical squamous morphology of the A549 line, showing elongation and rapid multiplication. Cells achieved 70–80% confluency between the 24 and 48 h time points. At the final time point, a few rounded, detached cells could be observed in all the samples (including glass), which is typical of the culture reaching stationary/declining growth stages. In addition to routine microscopic observations of cellular morphology, we chose to perform calcein-AM staining to ensure that no cell death was happening at a prominent scale. Calcein-AM staining performed at 48 h showed that most of the cells were viable in all the samples. One hot-press-molded PMP sample was treated with oxygen plasma prior to poly-L-lysine coating. This slightly improved cell adherence, especially within the first 24 h, as may be seen in Appendix A.

Due to the wavy nature of the extruded film PMP-E-20 (see Section 3.1), only some of the cells within the field of view were in focus simultaneously, as clearly seen in the Calcein-AM image and also apparent in the transmission light images (Figure 5). However, apart from that, all the PMP prototypes supported the transmission light and fluorescence microscopy imaging of cells (Figure 5 and Figure 6).

This assessment of cell behavior on the PMP prototypes compared to glass was performed in Non-sensor devices. In these devices, the culture chambers were open like in standard well plates; hence, there was oxygen supply by diffusion through the culture medium from the air–medium interface. This was carried out to reduce potential effects of differing oxygen concentration during the assessment, to enable a credible evaluation of the potential effect of the material and prototyping method. As the devices with PMP films will have oxygen diffusion both through the PMP films and culture medium, while the devices with glass will only have diffusion through the medium, we cannot be certain that the oxygen concentrations were the same for all variants. However, this method is highly likely to provide more similar oxygen concentrations than using Sensor devices and was thus the best option identified to perform this evaluation.

## 4. Discussion

### 4.1. Opportunities in Using PMP for Modulating Oxygen Levels in OoC

When fabricating microfluidic devices for OoC applications, whether it is prototyping or mass production, the choice of fabrication technology depends on the device thickness requirements, since different fabrication methods are compatible with different thicknesses. Therefore, this work was set to investigate the effect of PMP film thickness on the oxygen concentration inside the cell culture chamber. When testing PMP films of thicknesses 20, 125, 133, 356, and 653 µm, the resulting oxygen concentration after 4 days of A549 cell culture was between approximately 15.6% and 11.6%. When plotting oxygen concentration after 4 days vs. film thickness, the data followed a linear trend. With the limited data available, we could not identify any secondary effect of film prototyping method on the oxygen concentration, i.e., an effect on permeability via microstructure.

An exact prediction of the oxygen concentration inside such a device is difficult. The oxygen demand inside the device is determined by the cells’ oxygen consumption rate (mol cell^−1^ s^−1^) and the number of cells per area, giving an oxygen consumption rate in the unit mol cm^−2^ s^−1^. The supply of oxygen is by diffusion from the surrounding air through the PMP film. This is described by the following expression based on Fick’s first law for gas flux (*F*) through a film:(1)$$F=P*\;\frac{\Delta p}{x}$$
where *P* is the permeability of the gas through the polymer, Δ*p* is the partial pressure difference over the film, and *x* is the thickness of the film. The unit of *F* is mol cm^−2^ s^−1^. The permeability is constant for a certain gas and film material (assuming no effect of film prototyping method). A complicating factor for predicting the oxygen concentration is that the oxygen pressure difference across the film is created by the cells’ oxygen consumption inside the device, which is a dynamic factor [12]. For a deeper understanding of the dynamics of oxygen delivery to cells in culture, the reader is referred to the excellent review by Place et al. [12]. In brief, according to Wagner et al. [53], mammalian cell oxygen consumption rates may vary from <1 to >350 mol cell^−1^ s^−1^. It depends on the cell type and function and its biological and metabolic state, which again is influenced by the oxygen concentration through oxygen-utilizing enzymes like the hypoxia-inducible factor prolyl-hydroxylases (HIF-PHDs) and factor inhibiting HIF (FIH) [12,53]. Also, as the cells grow from sparsely seeded towards confluency, the number of cells and, thereby, cellular oxygen consumption rates increase [12]. Therefore, integrating sensors in the device to monitor the resulting oxygen concentration is a useful tool.

The flux of oxygen from ambient air through the PMP film into the Sensor devices in this work is analogous to the flux of oxygen through the culture medium in standard in vitro adherent cell culture in polystyrene well plates. There, the ambient oxygen pressure creates an equilibrated oxygen concentration in the medium at the air–medium interface, and as the adherent cells at the bottom of the wells consume oxygen, an oxygen concentration gradient is created from the air–medium interface to the cells at the bottom of the wells [12]. If too much medium is added to the wells (increasing the effective thickness for the permeation), the cells may experience hypoxic conditions [12]. Therefore, standard well plates have recommended volume ranges. A unique opportunity, if utilizing PMP as the sole material for an OoC device, is the possibility of spatially uniform oxygen supply by oxygen diffusion through all device walls. In gas-permeable OoC devices made with PDMS, the PDMS is usually bonded to glass, due to, amongst other things, the low mechanical robustness of PDMS [34]. This eliminates the oxygen supply through one side of the device.

For OoC models that seek to imitate in vivo functionality, replicating the in vivo oxygen conditions is essential [12]. At physiological conditions, the oxygen concentration in the human body ranges from 13–14% in arterial blood to 5–8% in venous blood, lung, liver, kidney, placenta, and bone marrow, to 4% in the brain, 3% in skeletal muscle, and 0.5–4.5% in lymphoid organs [13,14]. As the PMP films in this study, with various thicknesses compatible with different fabrication methods, showed oxygen levels of 11.6–15.6% with 2D culture of adherent cells, this suggests that PMP may in fact facilitate the relevant oxygen levels in OoC models. By adjusting the thickness of the device walls and/or the ambient oxygen partial pressure, we find it likely that the exact oxygen requirement for most 3D organ models may be attained. By continuously monitoring the oxygen concentration, it would also be possible to modulate the oxygen concentration in the model during an experiment by adjusting the ambient oxygen partial pressure.

### 4.2. Cell Culture Compatibility

As the prototyping method may affect the surface morphology and roughness of thermoplastic films, and it is known that these factors may affect cell adherence [54,55], the cell adherence and growth of A549 cells were compared on all prototyped PMP films and glass coverslips. All samples were coated with poly-L-lysine prior to cell seeding, as this is the established standard procedure for glass coverslips in the lab. Similar results for cell adherence, morphology, proliferation, and viability were found for all samples, indicating that coating with poly-L-lysine is adequate to enable cell adherence, and it potentially reduces the effects of surface morphology. A similar result has been reported earlier for a polished PMP film [30], although it was polished with a diamond polish instead of a paste intended for acrylics as in this study. Nishikawa et al. [33] and Danoy et al. [44] utilized PMP sheets attached to a 24-well plate format for the culture of hepatocytes and coated the PMP with collagen type I-P before cell seeding. Taken together, this suggests that standard procedures for coating microscope slides, coverslips, and polystyrene well plates may be equally successful for increasing cell adherence to PMP.

An alternative method for increasing cell adherence to PMP is to perform plasma treatment [54,55]. Therefore, we also tested PMP films first treated with plasma and then poly-L-lysine and found a slight increase in cell adherence in the first 24 h. However, due to the additional time and equipment needed to perform plasma treatment, combined with the small experimental effect, the procedure involving only poly-L-lysine coating was chosen for the main experiments. However, plasma treatment may be a viable approach when dealing with poorly adhering cell lines or in scenarios when using attachment agents such as poly-L-lysine is not desirable.

### 4.3. Comparison of Prototyping Methods for Manufacturing Transparent PMP Films

In this work, the devices had a chamber structure prototyped by milling in PC and a bottom layer of optically transparent PMP film, prototyped by three different methods. The results of prototyping studies using these three methods are compared below.

Of the three film prototyping methods tested, our preferred method for OoC applications was the hot-press molding method. The transparency of the films was good and reproducible, and the equipment is simple and available in many labs. The drawback of hot-press molding is the long time needed to make a film.

When the process parameters are identified, the hot-press molding procedure is easy to perform, and it allows for prototyping different film thicknesses by exchanging the shims. However, in the experimental set-up used herein, the shims were not attached to the glass plates, and therefore, the shims moved during molding. This made it difficult to predict the resulting film thickness. For future studies, improving the set-up with fixed parts will reduce the workload for optimizing parameters. Using a commercial film pressing tool (e.g., the Atlas Constant Thickness Film Maker from Specac, Orpington, UK) will also be considered.

Regarding costs and the need for equipment, this prototyping method requires access to a small hot press, which is rather inexpensive. Little PMP material is wasted in this process, thereby reducing the material costs. We recommend this prototyping method if small quantities of transparent PMP films are required and if the application requires a range of film thicknesses and/or thicknesses which are not commercially available. A benefit of this method is that it may also be applied for the prototyping of structured PMP parts. This is sometimes referred to as hot embossing when a film or sheet is softened by heating and then structured by pressing. Compared to injection molding, the 3D geometrical freedom of hot-press molding is limited. On the other hand, hot-press molding is better suited for making thin-walled parts.

The second prototyping method in this study, film extrusion, is an established method for making high quality films with a wide range of thickness. Film extrusion can produce films on an industrial scale, while lab-scale film extrusion lines as used in this study are suitable for prototyping. The 20 µm film extruded in this study had some waviness due to process instabilities. Hence, it was not optimal for the microscopy of cells, as only a portion of the cells were in focus for a specific field of view and focus depth. In this work, we wanted to push the lower limit regarding film thickness, to evaluate the oxygen delivery to cells for a wide range of film thicknesses. Extruded PMP films with thickness down to at least 50 µm are available commercially (although typically with one frosted side). Hence, we believe it is feasible to produce good “flat” PMP films with thickness down to about 20 µm with lab-scale extrusion lines, after optimizing the process parameters and perhaps selecting a PMP grade with more appropriate rheology. Ma et al. [38] produced PMP films with thicknesses in the range 22–80 µm by cast film extrusion (the extrusion process in our study). With a similar process, Yin et al. [39] produced PMP films with thicknesses in the range 15–50 µm. Johnson and Wilkes [40] produced 25 µm thick PMP films by blown film extrusion. Merkel et al. [41] also used blown film extrusion and produced 110 µm thick PMP films. With blends of low density polyethylene (LDPE) with 15 wt% PMP, they produced 8 µm thick films.

We recommend film extrusion for prototyping PMP for OoC devices if the goal is to fabricate relatively large quantities of film of the same thickness and without structure. (Structured films can also be made by using structured rolls). The drawback is that an extrusion line is needed. Such equipment is rather expensive and less common in labs than, e.g., hot presses.

The third prototyping method in this study was to mechanically polish the rough surface of a commercial non-transparent PMP film. Although the polished film was not as transparent as glass coverslips but rather had remnants of the texture of the original film in addition to some small dents, this did not pose a problem for the transmission light and fluorescence imaging of cells on the film. In this study, the polishing was performed using an “acrylic polishing kit”. In a previous study [30], we polished the same commercial PMP film with diamond paste (grit size down to 1 µm). The diamond polishing resulted in occasional scratches and bending of the PMP film, while this was avoided when using the polishing kit meant for acrylics. The polishing procedure is relatively time-consuming and labor-intensive. However, it does not require special equipment and appears to be the most available method for making transparent PMP films. We recommend this method of prototyping if only small quantities of transparent PMP film are needed, if the required film thickness for the application matches that of commercially available films, and—as for film extrusion—if the goal is to manufacture only non-structured films. Future work may explore polishing via heat, hot-press molding, or chemicals.

### 4.4. Microstructure of the PMP Films

The gas permeability of a PMP film is generally affected by the PMP microstructure. Yoshimizu and Okumura [52] reported that the permeability of gas molecules such as O_2_ and CO_2_ through PMP films decreased with increasing crystallinity and increasing molecular orientation in the film. In their study, undrawn and drawn films were compared. The drawn films had dichroic ratios (based on the 918 cm^−1^ band) in the range 2.3 to 2.6 for draw ratios in the range 2 to 5. The undrawn film had a dichroic ratio below 1.5 (estimated from the published spectra; the actual value was not reported). Johnson and Wilkes reported a dichroic ratio of 2.7 for a PMP film made by blown film extrusion [51]. In our study, the highest dichroic ratios for this band were 1.6 and 1.8 for PMP-P-125 and PMP-E-20, respectively (Figure 2). Hence, the permeabilities through these two extruded films may be somewhat lower than through the hot-press-molded films. However, the two films with almost the same thickness but very different dichroic ratios (PMP-P-125 and PMP-M-133) gave almost the same O_2_ concentrations in the chambers (Figure 4). In fact, PMP-P-125, which is slightly thinner and has a larger dichroic ratio, gave a slightly higher O_2_ concentration. Hence, the oriented microstructure in PMP-P-125 does not seem to affect the permeability significantly.

The films in this study may also differ with regard to the degree of crystallinity, as well as other microstructure features. For the hot-press-molded films, the cooling rate determines the degree of crystallinity, and thinner films experience a higher effective cooling rate. For extruded films, the initial cooling rate is typically lower than in a molding process, because of the heat transfer to air, although an “air knife” is often used to enhance the cooling [38]. However, in addition to temperature-induced crystallization, extrusion processes may also cause orientation-induced crystallization. Ma et al. [38] measured the crystallinity of PMP films made by blown film extrusion. The films had thicknesses in the range 22–80 μm (obtained by varying the draw ratio with a constant die opening), and the crystallinity values were in a rather narrow interval (61.8 to 65.7%). The crystallinity of PMP films will be the topic of a separate study of overall PMP film microstructure. In addition to methods such as wide-angle x-ray scattering and calorimetry, the crystallinity may be assessed by IR spectroscopy. The PMP literature has not assigned specific IR bands to crystalline or amorphous (or mixed) phases, although Ma et al. [38] used the 918 cm^−1^ band to represent the crystalline phase. If we normalize the amplitudes of the bands in Figure 2 with the 918 cm^−1^ amplitude, some trends are observed for certain bands, indicating that these bands have a different phase assignment than the 918 cm^−1^ band. This may also be the reason why some bands show deviating trends between the films, e.g., the 954 cm^−1^ band for PMP-P-125 vs. PMP-E-20 (Figure 2).

## 5. Conclusions

In this paper, we reported on important aspects of OoC device prototyping in PMP. This research enables more informed design and fabrication decisions, as well as progression from lab-scale prototyping to production. Most importantly, as different fabrication methods are compatible with different device wall thicknesses, we investigated the effect of PMP wall thickness on oxygen concentration inside cell culture devices. The tested PMP thicknesses (20, 125, 133, 356, and 653 µm) provided oxygen concentrations from 15.6% to 11.6% after 4 days of cell culture, as opposed to glass which resulted in 3% oxygen. This indicates that PMP may facilitate the replication of in vivo oxygen concentrations in OoC models with all the tested PMP thicknesses and prototyping methods. The investigated thickness range of 20–653 µm covers typical fabrication techniques for thermoplastics, such as extrusion, hot-press molding, hot embossing, and injection molding.

We also showed that all three tested prototyping methods—extrusion, hot-press molding, and polishing a commercial, hazy film—were successful in achieving adequate transparency to support the transmission light imaging of adherent cells. We discussed in which situations each method is recommended, as there is not one method suitable for all applications. For our application, hot-press molding was concluded as the preferred film prototyping method, due to excellent and reproducible film transparency, the possibility to easily vary film thickness, and the equipment being commonly available.

Via microscopy imaging, we also showed that all the PMP films gave similar A549 cell adherence, proliferation, and viability as glass coverslips (all samples coated with poly-L-lysine).

Molecular orientation in the PMP films was investigated to assess the possible effects of the prototyping method and molecular orientation on oxygen permeability. As expected, the extruded film and the polished (also extruded) film showed clear orientation (PMP-E-20, PMP-P-125), while the thick hot-press-molded films did not (PMP-M-356, PMP-M-653). Unexpectedly, the thinnest hot-press-molded film (PMP-M-133) had some orientation. Considering the similar oxygen levels for the two films with almost similar thickness but different degrees of orientation (PMP-P-125 and PMP-M-133), we conclude that the orientation is not a critical factor for the permeability of the films in this study.

## 6. Outlook

The results presented herein, combined with previous works, demonstrate that PMP supports cell experiments and microscopy [30,33,44,54,55]. This opens up interesting possibilities which this material can offer within the OoC field. In particular, the oxygen permeability of PMP allows for oxygen concentrations relevant for in vivo human organs. Furthermore, PMP is suitable for mass production, and hence, it is a unique candidate for the industrial application of OoC technology. Further developments to make the prototyping of transparent PMP films and parts more easily accessible to the OoC community could enhance the speed of the adoption of this material in research. We believe that PMP is an important material which can enable unique OoC models in the years to come.

## Figures and Tables

**Figure 1 micromachines-15-00898-f001:**
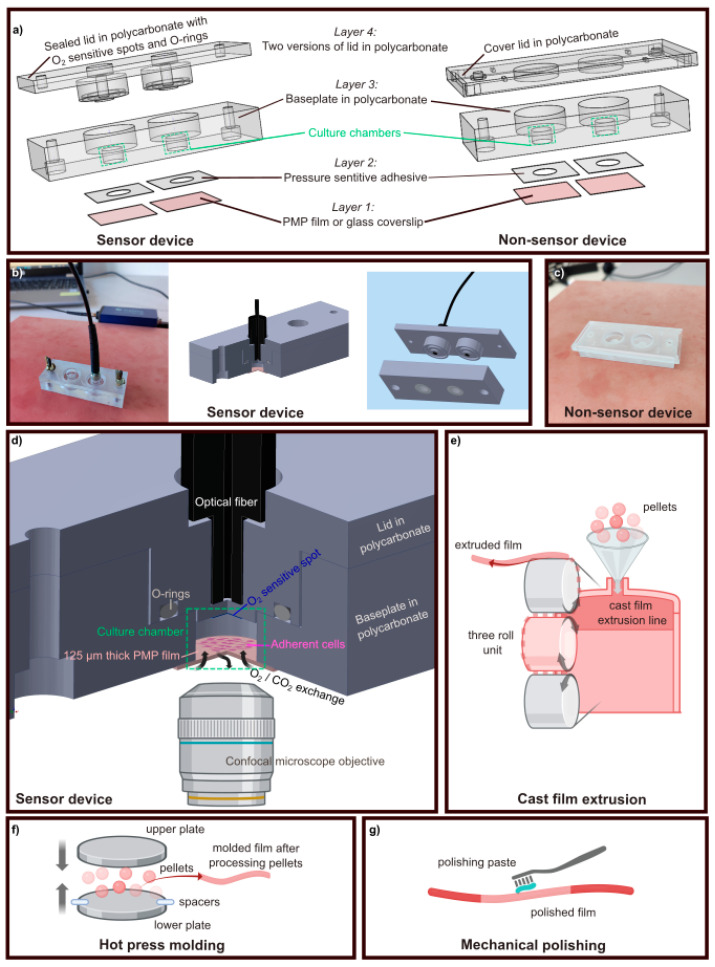
The culture chamber devices and PMP film prototyping: (**a**) Depiction of the four layers that make up the Sensor device (left) and Non-sensor device (right). Layers 1–3 are the same for both devices, while layer 4—the lid—ensures a tight seal and oxygen sensing in the Sensor device, while only constituting a support cover to ensure sterility in the Non-sensor device. (**b**) Photograph and schematics of the Sensor device with the optical fiber used to read out the oxygen concentration inserted into the customized cavity. (**c**) Photograph of the Non-sensor device. (**d**) A zoom-in on a culture chamber in the Sensor device, showing the sealed chamber with adherent cells growing on the gas-permeable PMP film, the O_2_ sensitive spot, O-rings for sealing, and the optical fiber for read-out of O_2_ concentration inside the chamber. (**e**–**g**) General schematics of prototyping methods tested for fabrication of transparent PMP films: (**e**) cast film extrusion, (**f**) hot-press molding, (**g**) mechanical polishing. Inventor 2022 was used for creation of the 3D device images. BioRender.com (accessed on 21 November 2022, 21 June 2024) was used for illustrations of microscope objective, adherent cells, and PMP film prototyping.

**Figure 2 micromachines-15-00898-f002:**
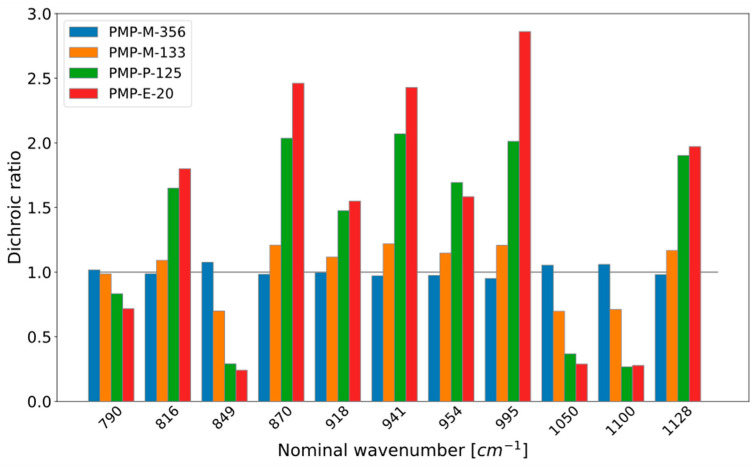
IR dichroism data for four of the PMP films. The gray horizontal bar indicates a ratio of 1.

**Figure 3 micromachines-15-00898-f003:**
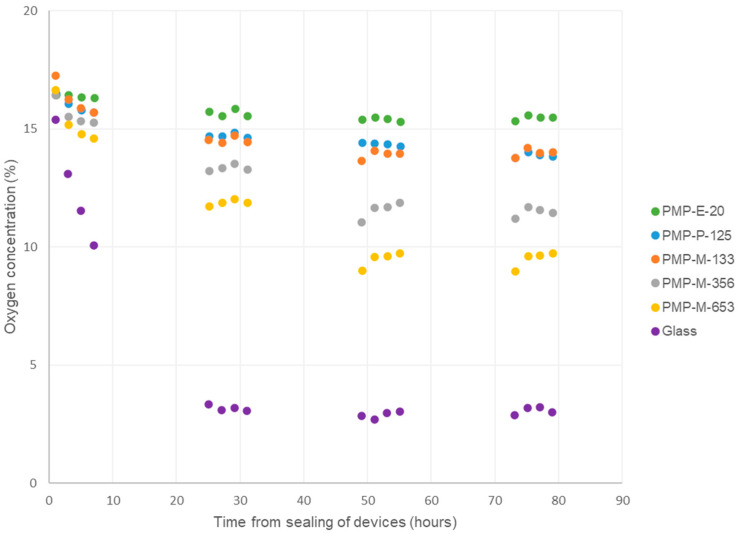
Monitoring of oxygen concentration inside cell culture chambers in Sensor devices, with different PMP prototypes or glass coverslip as layer 1, i.e., the cell-adhering layer. A549 cells were cultured inside the sealed devices for four days. *n* = 1.

**Figure 4 micromachines-15-00898-f004:**
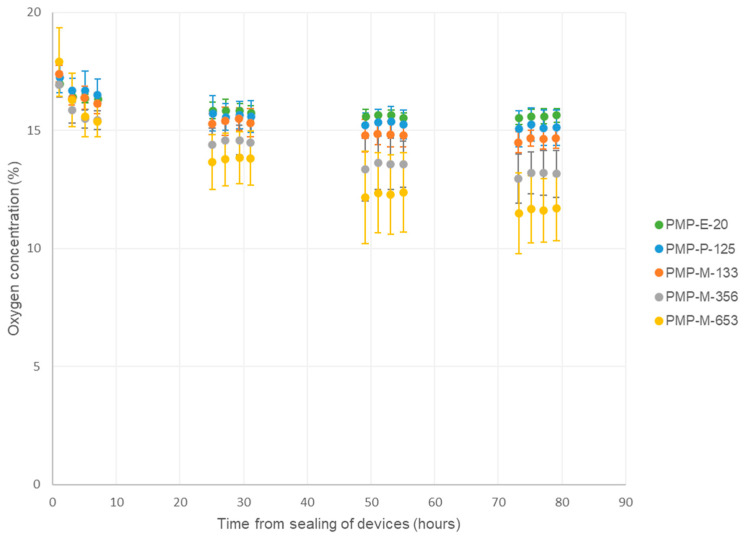
Monitoring of oxygen concentration inside cell culture chambers in Sensor devices, with different PMP prototypes as layer 1, i.e., the cell-adhering layer. A549 cells were cultured inside the sealed devices for four days. Data are presented as average ± SD, *n* = 4.

**Figure 5 micromachines-15-00898-f005:**
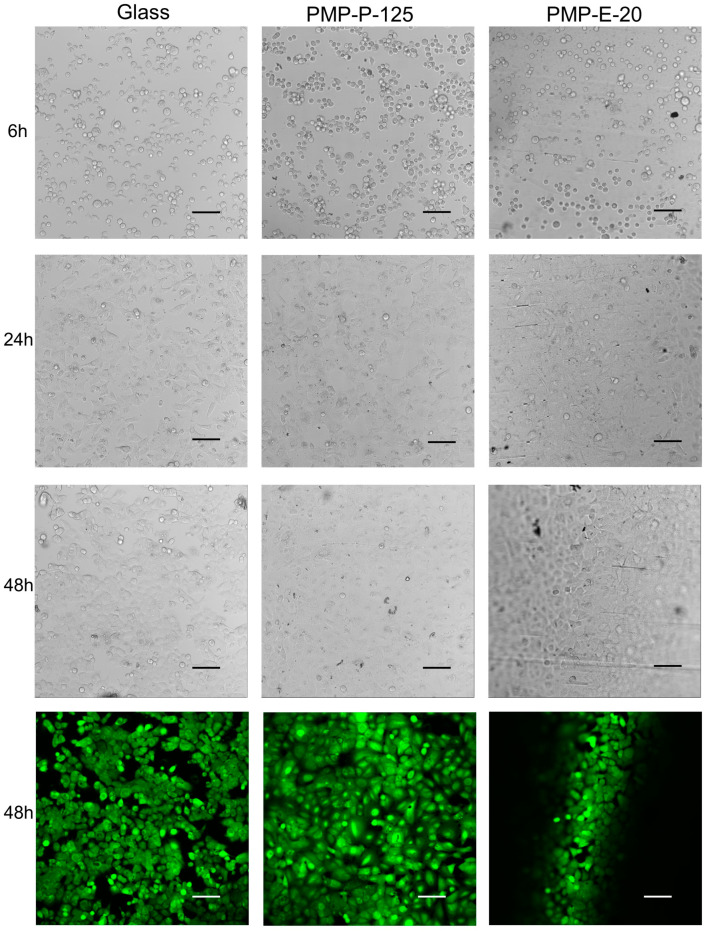
Microscopy images showing cell adherence, proliferation, and viability of A549 cells cultured in Non-sensor devices with either glass coverslips, PMP-P-125 films or PMP-E-20 films as layer 1 in the device, i.e., the cell-adhering layer. All materials were coated with poly-L-lysine prior to cell seeding. Transmission light images at 6, 24, and 48 h follow the cell growth, while the green fluorescence at 48 h display calcein-AM staining, where green indicates living cells. Due to the wavy nature of PMP-E-20, only some cells are in focus. Scale bar: 100 µm.

**Figure 6 micromachines-15-00898-f006:**
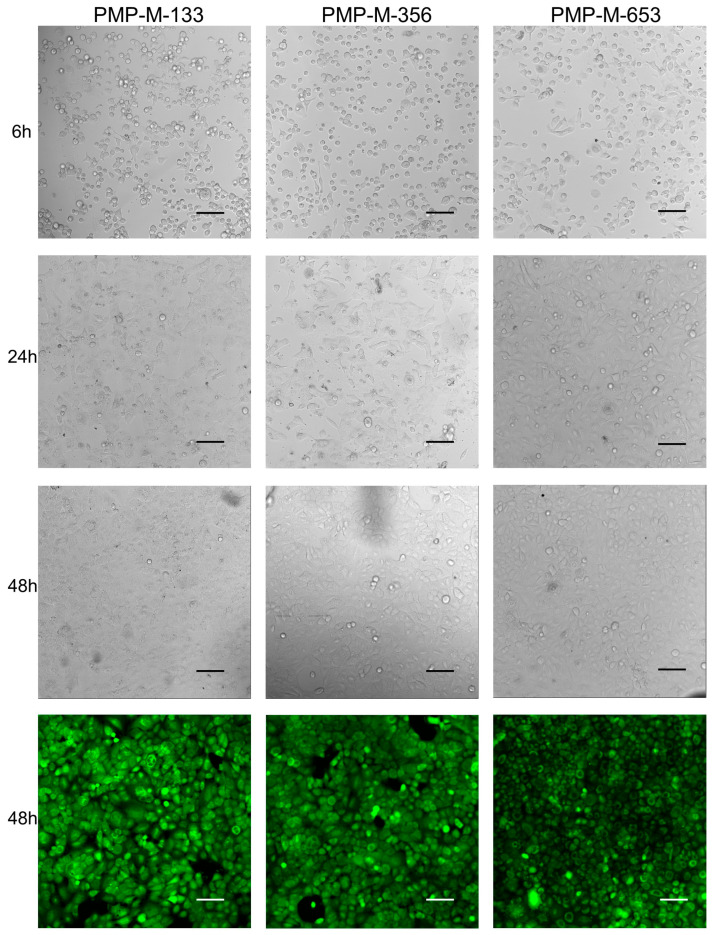
Microscopy images showing cell adherence, proliferation, and viability of A549 cells cultured in Non-sensor devices with either PMP-M-133, PMP-M-356, or PMP-M-653 films as layer 1 in the device, i.e., the cell-adhering layer. All materials were coated with poly-L-lysine prior to cell seeding. Transmission light images at 6, 24, and 48 h follow the cell growth, while the green fluorescence at 48 h display calcein-AM staining, where green indicates living cells. Scale bar: 100 µm.

**Table 1 micromachines-15-00898-t001:** Overview of the PMP films used in this work, listing prototyping method, thickness, and sample name.

Prototyping Method	Thickness	Sample Name
Extrusion	20 µm	PMP-E-20
Polishing commercial film	125 µm	PMP-P-125
Hot-press molding	133 µm	PMP-M-133
Hot-press molding	356 µm	PMP-M-356
Hot-press molding	653 µm	PMP-M-653

## Data Availability

Data are contained within the article and Appendix A.

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
