# Peer review of "Prototyping in Polymethylpentene to Enable Oxygen-Permeable On-a-Chip Cell Culture and Organ-on-a-Chip Devices Suitable for Microscopy"

_micromachines, 2024, doi:10.3390/mi15070898_

Round 1

Reviewer 1 Report

Comments and Suggestions for Authors

Attached

Reviewer 2 Report

Comments and Suggestions for Authors

The article entitles “Prototyping in Polymethylpentene to Enable Oxygen Permeable on-a-Chip Cell Culture and Organ-on-a-Chip Devices Suitable for Microscopy” is a continuation of the article titled “Application of Polymethylpentene, an Oxygen Permeable Thermoplastic, for Long-Term on-a-Chip Cell Culture and Organ-on-a-Chip Devices” (Micromachines (Basel) 2023;14:532. https://doi.org/10.3390/MI14030532/S1) published in this journal last year, 2023, as indicated by the authors.  In this work, the permeable nature of the material is explored based on how the sheets are obtained and their thickness. To this end, oxygen permeability has been measured in chip cultures, and cell viability, proliferation, and adhesion have been studied.

It is a very descriptive article that provides a detailed narrative but is supported by limited data. The thorough descriptions offer a clear understanding of the subject matter, yet the scarcity of data may weaken the overall strength of the conclusions. Additional empirical evidence would enhance the credibility and robustness of the findings presented.

In my opinion, it is not a very original article considering the previous one. In fact, the same work scheme has been practically reproduced according to methodology, introduction and discusion. Furthermore, the data presented alone lack the necessary relevance to constitute an article on their own. Therefore, I would invite the authors to reconsider the work and offer a different approach.

Analysing each of the sections:

1.       Abstract

a.       In general, it is a well-written summary that captures all the information from the work.

b.      Line 19: specify which cells are the A549 (cancerous lung epithelial cells).

2.       Introduction

Very similar to that presented in the previous article. It detracts from its novelty and value.

Complete with recent references

3.       Materials and methods

a.       The methodology employed is well described. However, I consider that the methodology must be improved

b.      Line 48: use the acronym "OOC"

c.       Figure 1: some photographs were used in your previous article (Micromachines, 2023). Although the images can be used, new ones should be considered. In my opinion, it would make a better impression.

4.       Results

a.       Figure 2. Images must be improved; the quality is low. I consider that the visual perception of transparency is not very rigorous. In this case, the transmittance determination must be taken into account.

b.      Figure 3: I don´t understand why this graph is included.

c.       Figure 4. Incorporate the control data (glass) and the sample size should be increased in all experiments. Additionally, a statistical analysis of the data should be conducted.

d.      To complete the cell viability study, a staining to detect dead cells (Propidium Iodide) should be included, accompanied by an analysis of the data, presenting both data and statistical analysis.

e.      Figure 5-6. It would be necessary to include some microscopy images where the cells can be better observed by increasing the magnification. Additionally, in the case of the PMP-E-20 sample, so that the cells can be seen.

5.       Discussion

                      There is a lack of empirical data to support the provided information

Comments on the Quality of English Language

 Moderate editing of English language required

Reviewer 3 Report

Comments and Suggestions for Authors

The author explored the use of polymethylpentene (PMP) material-based microfluidic devices for simulating organ-on-chip models, with a specific focus on their performance in terms of oxygen permeability. The article demonstrated the potential of PMP in regulating the microenvironment by examining how various thicknesses of PMP films affected oxygen concentration in cell cultures. This is significant for enhancing the biocompatibility and experimental outcomes of organ-on-chip technologies. Please find my comments below:

  1. In Figure 5, the Calcein-AM staining on PMP-E-20 films revealed a significantly brighter central area compared to the edges. What factors may have contributed to this difference in fluorescence distribution?

  2. While Calcein-AM is a widely used fluorescent dye for labeling live cells due to its enzymatic cleavage in live cells, releasing green fluorescence to indicate the location and number of live cells, it does not differentiate between live and dead cells. For a more comprehensive assessment of cell viability, it is advisable to combine Calcein-AM with PI probes to distinguish between live and dead cells. Additionally, more references related to cell microenvironment control are necessary. Consider including the following: (The Innovation, 2022, 3(5), 100294; The Innovation, 2022, 3(6), 100313).

  3. The article lacks breadth and depth in its incorporation of recent literature citations. To enhance the coverage of relevant literature, consider including references related to chip fabrication design, such as (The Innovation, 2022, 3(5), 100282; The Innovation, 2023, 4(4), 100447).

Round 2

Reviewer 1 Report

Comments and Suggestions for Authors

The authors have addressed my comments well and have improved the quality of the manuscript.

Reviewer 2 Report

Comments and Suggestions for Authors

I greatly appreciate the answers provided by the authors and I am aware of the effort involved in preparing a work, but I consider that the shortcomings and weaknesses of this work have not been resolved, especially the experimental ones. Therefore, I continue to believe that the work does not meet the minimum requirements for admission.

Comments on the Quality of English Language

-

Reviewer 3 Report

Comments and Suggestions for Authors

The paper address the comments and should be accept.
